# Autoregressive Unsupervised Image Segmentation

## Reproducibility Summary

**Scope of Reproducibility**

Ouali *et. al.*[1] consider the problem of unsupervised image segmentation, that is, the assignment of a class label to each pixel of an input image without the use of any training labels. The authors claim a novel method for performing this task, which involves training a convolutional neural network by maximizing the mutual information between outputs obtained using different orderings of the input image. The paper reports state-of-the-art pixel accuracy for unsupervised methods on the COCO-Stuff and Potsdam benchmark datasets, as well as on 3-class variants of these datasets, Potsdam-3 and COCO-stuff-3. The scope of this reproduction is to create an implementation of the described method and verify its performance on the benchmark datasets.

**Methodology**

We created an original implementation of the described method using the PyTorch framework. All experiments were conducted on a single desktop computer with an Nvidia GTX 1080Ti GPU. The total compute budget was approximately 40 GPU hours.

**Results**

We reproduced the accuracy claimed in the paper to within 1% on the Potsdam-3 dataset and to within 4% on the Potsdam dataset. Additionally, we found that the inclusion of a self-attention layer can improve model performance as reported in the paper. However, our model's accuracy on the COCO-Stuff dataset is drastically lower than is reported in the paper. This may be due to the smaller model and reduced batch size that we adopted in our reproduction due to limited computational resources.

**What was easy**

The model architecture is easily implemented using standard machine learning frameworks. The main building block of the model is the masked convolution, which can be implemented as a simple extension to regular 2D convolutions. Additionally, this work builds on a previous paper which uses the mutual information loss as an unsupervised clustering objective [2]. The published code for the loss function from this paper was adapted to our application with minimal changes.

**What was difficult**

The model configurations specified by the authors are too large for single-GPU training. We used a smaller network and reduced batch sizes in our reproductions, but we note that this is may lead to differences in model performance.

**Communication with original authors**

We corresponded by email with the original authors to clarify the architectural details of the self-attention layer in the model.

Preprint. Under review.

# 1 Introduction

Image segmentation is the process of assigning a class label to each pixel of an input image. This paper presents a novel unsupervised approach to image segmentation called Autoregressive Clustering (AC). AC builds on previous techniques which perform segmentation or clustering using a mutual information (MI) objective. The main idea behind these approaches is to maximize the MI between different constructed views of the inputs [3]. Recent works have employed various methods for constructing different views, for example using geometric transformations [2]. In this paper, the views are taken to be autoregressive orderings over the input pixels of the image.

The proposed training process is shown in figure 1. An input image $x$ is passed through an autoregressive model $\mathcal{F}$ using an ordering $o_i$, resulting in a segmentation $y = \mathcal{F}(x; o_i)$. The autoregressive model uses masked convolutions to enforce causality over the inputs, ensuring that the output at any pixel depends only on the input pixels which precede it in the ordering $o_i$. This process is then repeated with a second ordering $o_j$ to produce a segmentation $y' = \mathcal{F}(x; o_j)$. The model weights are then updated to maximize the mutual information objective function $I(y, y')$. During inference, no masking is applied to the convolution weights of the model, and the segmentation result is obtained directly from the model output.

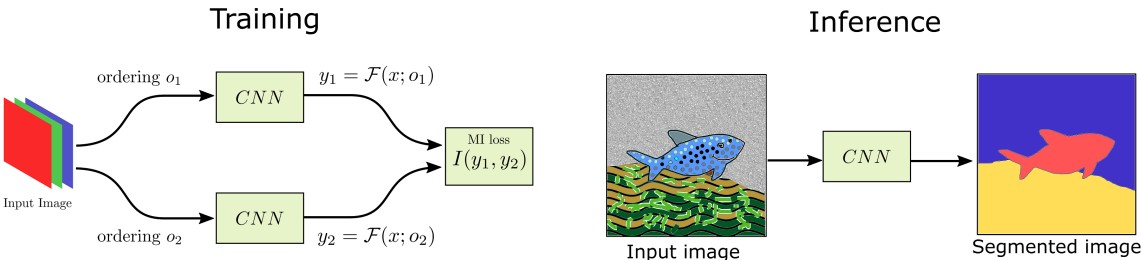

Figure 1: A diagram of the autoregressive image segmentation process.

# 2 Scope of Reproducibility

## 2.1 Target Claims

We chose to test the claims related to Autoregressive Clustering (AC) which had quantifiable results. Under this criterion the main claims of the paper can be summarized as follows:

1. The proposed method shows 66.5% accuracy on the Potsdam-3 dataset. This claim is supported by our results shown in table 5 in section 4.1.

2. The proposed method shows 47.9% accuracy on the Potsdam dataset. This claim is supported by our results shown in table 6 in section 4.2.

3. The proposed method shows 72.9% accuracy on the COCO-Stuff-3 dataset. We do not find evidence supporting this claim with our model. Our relevant results are shown in table 7 in section 4.3.

4. The proposed method shows 30.8% accuracy on the COCO-Stuff dataset. We do not find evidence supporting this claim with our model. Our relevant results are shown in table 8 in section 4.4.

5. The addition of a self-attention layer in the model increases its accuracy over models where no self-attention is present. This claim is supported by our results shown in tables 5 and 6.

Due to the differences in the training conditions between the original paper and our recreation, we did not expect identical numerical results. Instead, we applied our judgement to determine which claims were supported by our evidence, taking into account the limitations of our approach. Detailed discussion on this point is provided in section 5.

## 2.2 Untested Claims

The paper contains a number of separate contributions. While we feel that our reproduction captures the crucial elements of the method, there are additional aspects which remain untested:

1. The paper presents two different learning objectives, autoregressive clustering (AC) and autoregressive representation learning (ARL). AC seeks to obtain outputs over a discrete set of classes, whereas ARL generates representations of the input within a continuous output space. We restrict our attention to AC as it is reported to perform better than ARL and is used in the majority of the authors' experiments.

2. The authors extend the number of autoregressive orderings from 8 to 16 by including zig-zag orderings. The paper demonstrates how the zig-zag orderings can be obtained via masking in a self-attention block, however it remains unclear how they can be applied to the convolutional parts of the network using masked convolutions. As such, we omit the zig-zag orderings in our experiments, and compare our results only to those in the paper that are obtained without them.

## 3 Methodology

We created an original implementation of the method based on the description provided in the paper and the supplemental material. The code for the loss function and the data preparation scripts were adapted from published code associated with a previous paper on unsupervised image segmentation [2]. All experiments were conducted on a single desktop computer with an Nvidia GTX 1080Ti GPU.

### 3.1 Model descriptions

The AC model $\mathcal{F}$ is a fully convolutional neural network comprising three different components as shown in figure 2. The first is an encoder $h$, followed by an autoregressive encoder $g_{ar}$ and a decoder $d$. The layers which make up the encoder $h$ and the decoder $d$ are specified in tables 4 and 5 in the supplementary material of the original paper. The autoregressive encoder $g_{ar}$ is composed of a number of residual blocks [4], each comprising the layers specified in table 7 of the supplemental material in the original paper. The autoregressive orderings are enforced in $g_{ar}$ by replacing the regular convolutions with masked convolutions. If a self-attention mechanism is used, it is placed between the second and third residual layers of $g_{ar}$.

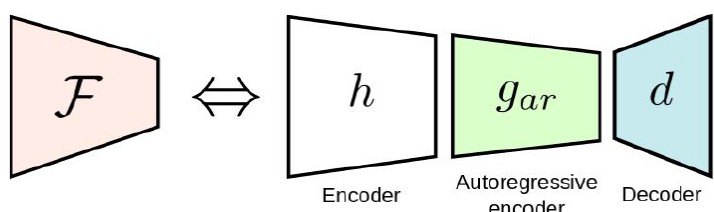

Figure 2: Block diagram of the AC model architecture. Reproduced from
`https://yassouali.github.io/autoreg_seg/data/slides_autoreg_seg_ECCV20.pdf`

In the majority of their experiments, the authors employed autoregressive encoders $g_{ar}$ with 5 residual blocks. Since the number of convolutional filters is doubled with each successive residual block, this results in a large model with over 50 million trainable parameters. To allow for training on a single GPU, we reduced the number of residual blocks to 4 for our experiments. The number of trainable parameters in each model is summarized in table 1.

| Num. Res. Blocks | Attention | Trainable Parameters |
|:---:|:---:|:---:|
| 4 | No | 13.3M |
| 4 | Yes | 13.5M |
| 5 | No | 53.1M |
| 5 | Yes | 53.4M |

Table 1: Summary of model architectures used for AC. Due to limited computational resources we only test models with 4 residual layers in this reproduction.

### 3.2 Datasets

#### 3.2.1 Potsdam

The Potsdam dataset [5] is composed of high-resolution RGBIR aerial images taken over the city of Potsdam in Germany. The images are accompanied by pixel-level segmentation masks where each pixel is labeled as one of six different classes: roads, cars, vegetation, trees, buildings, and clutter. The authors experiment both with the original labels, and with a 3-label variant called Potsdam-3 in which the roads and cars, vegetation and trees, and buildings and clutter classes are merged. The images are split into 200x200 sections for training.

A data request form for downloading the Potsdam dataset is available at `https://www2.isprs.org/commissions/comm2/wg4/benchmark/data-request-form/`.

#### 3.2.2 COCO-Stuff

The COCO-stuff dataset [6] comprises 164k images with pixel-level segmentation masks. The authors use two different versions of this dataset. In the first, they evaluate across the 15 coarse stuff labels from the original dataset. The second is CocoStuff-3, which contains 4 labels: ground, sky, plants, and other. Only the images where at least 75% of the pixels belong to the stuff classes were retained, and the images were resized to 128x128 before use.

Instructions for downloading the COCO-Stuff dataset are available at `https://github.com/nightrome/cocostuff`.

#### 3.2.3 Preprocessing

To ensure a fair comparison between our reproduction, the original paper, and previous works, we used the data preprocessing scripts published along with the code for IIC [2] to prepare the file lists for the training and validation of our model. A summary of the datasets is given in table 2.

| Dataset | Train Images | Test Images | Num. Classes | Resolution |
|---|---|---|---|---|
| Potsdam | 7695 | 855 | 6 | 200x200 |
| Potsdam-3 | 7695 | 855 | 3 | 200x200 |
| COCO-Stuff | 49269 | 2175 | 15 | 128x128 |
| COCO-Stuff-3 | 49269 | 2175 | 4 | 128x128 |

Table 2: Summary of the benchmark datasets used in this study.

### 3.3 Hyperparameters

We followed the hyperparameters presented in the paper, however we changed the batch size to accommodate our limited computational resources. Additionally, we obtained unstable results on the Potsdam dataset using the original learning rate of 4e-5. As a result, we reduced the learning rate to 1e-6. The Adam optimizer was used with $\beta_1 = 0.9$ and $\beta_2 = 0.999$. The model weights were initialized using Xavier initialization (using `torch.nn.init.xavier_normal_()`). The remainder of the hyperparameters are specified in Table 3. Note that more than one model configuration was used for the Potsdam and Potsdam-3 datasets, and the hyperparameters are specified for each.

| Dataset | Experiment No. | Attention | Learning Rate | Batch Size | Output Stride[a] | Displacements[b] |
|---|---|---|---|---|---|---|
| Potsdam-3 | model_001 | No | 1e-6 | 18 | 4 | 10 |
| | model_002 | Yes | 1e-6 | 10 | 4 | 10 |
| | model_003 | No | 1e-6 | 4 | 2 | 10 |
| Potsdam | model_004 | No | 1e-6 | 18 | 4 | 10 |
| | model_005 | Yes | 1e-6 | 10 | 4 | 10 |
| | model_006 | No | 1e-6 | 4 | 2 | 10 |
| COCO-Stuff-3 | model_007 | No | 4e-5 | 30 | 4 | 10 |
| COCO-Stuff | model_008 | No | 6e-6 | 30 | 4 | 10 |

[a] This refers to the size reduction of the input following the convolutional stem section of the network. See table 4 in the sup. mat. of the original paper for details.

[b] This refers to the number of displacements to be averaged over in the mutual information loss function.

Table 3: Hyperparameters used in our experiments.

### 3.4 Experimental setup and code

We used our original code to conduct all of our experiments. The code is available at `https://github.com/Max-Manning/autoregunsupseg`.

We evaluated the models using pixel accuracy. Following the authors, we used the Hungarian algorithm to compute the best one-to-one mapping between the output class labels and the ground truth labels prior to computing the pixel accuracy. Our code computes the pixel accuracy on the testing set after each training run. The commands used to run each of our experiments are listed in the README file. The experiments are numbered for easy reference with the contents of this report. All experiments were run for 10 epochs, and we report the best pixel accuracy from any epoch.

### 3.5 Computational requirements

All the experiments were conducted on a desktop computer with an Nvidia GTX 1080Ti GPU. The average training time for various configurations of the model are reported in table 4. At least 10GB of VRAM is required to perform the experiments.

| Dataset | Training time, 10 epochs (min.) |
|---|---|
| Potsdam | 80 |
| Potsdam-3 | 80 |
| COCO-Stuff | 210 |
| COCO-Stuff-3 | 210 |

Table 4: Summary of training time for each model used.

## 4 Results

The results of our experiments support claims 1 and 2, which relate to the model's performance on the Potsdam-3 and Potsdam datasets. However, we do not find evidence supporting claims 3 and 4, which relate to the model's performance on the COCO-Stuff-3 and COCO-Stuff datasets. In the following sections we present the results of each of our experiments in detail.

### 4.1 Performance on the Potsdam-3 dataset

A comparison of our results with the reported pixel accuracy on the Potsdam-3 dataset is shown in table 5. Despite the smaller model and reduced batch size, our results are comparable with the reported values. In particular, using a model

with an output stride of 4 and an attention layer, our result differs from the reported value by only 1%. Furthermore, we confirm that the inclusion of an attention layer improves the model performance over the baseline model. Finally, we note that reducing the output stride from 4 to 2 leads to increased performance as is reported in the paper.

| Source | | Model Description | | | Performance |
|---|---|---|---|---|---|
| Reference | Location | Num. res. layers | Attention | Output stride | Pixel Accuracy (%) |
| Original paper | Table 1(c), row 1 | 5 | No | 4 | 61.0 |
| | Table 1(c), row 2 | 5 | Yes | 4 | 66.3 |
| | Table 1(a), row 2 | 5 | Yes | 2 | 66.4 |
| Reference | Experiment No. | Num. res. layers | Attention | Output stride | Pixel Accuracy (%) |
| Ours | model_001 | 4 | No | 4 | 56.3 |
| | model_002 | 4 | Yes | 4 | 65.3 |
| | model_003 | 4 | No | 2 | 63.5 |

Table 5: Comparison of our results with the original paper for the Potsdam-3 dataset.

## 4.2 Performance on the Potsdam dataset

Our results for the Potsdam dataset are compared with those from the original paper in table 6. Without an attention layer and with an output stride of 4, our model achieves 6% lower pixel accuracy. The addition of an attention layer increases the performance, however the result remains 7% lower than the reported value. Interestingly, decreasing our model's output stride to 2 results in the best performance, bringing it to within 2.2% of the reported pixel accuracy.

| Source | | Model Description | | | Performance |
|---|---|---|---|---|---|
| Reference | Location | Num. res. layers | Attention | Output stride | Pixel Accuracy (%) |
| Original paper | Table 1(c), row 1 | 5 | No | 4 | 45.2 |
| | Table 1(c), row 2 | 5 | Yes | 4 | 47.9 |
| | Table 1(a), row 2 | 5 | Yes | 2 | 46.4 |
| Reference | Experiment No. | Num. res. layers | Attention | Output stride | Pixel Accuracy (%) |
| Ours | model_004 | 4 | No | 4 | 39.0 |
| | model_005 | 4 | Yes | 4 | 41.0 |
| | model_006 | 4 | No | 2 | 44.2 |

Table 6: Comparison of our results with the original paper for the Potsdam dataset.

## 4.3 Performance on the COCO-Stuff-3 dataset

We report a comparison of our results with the reported values on the COCO-Stuff-3 dataset in table 7. For this dataset our model significantly underperforms with respect to the original paper, with a pixel accuracy that is lower by 22%.

| Source | | Model Description | | | Performance |
|---|---|---|---|---|---|
| Reference | Location | Num. res. layers | Attention | Output stride | Pixel Accuracy (%) |
| Original paper | Table 3, col. 1 | 5 | No | 4 | 72.9 |
| Reference | Experiment No. | Num. res. layers | Attention | Output stride | Pixel Accuracy (%) |
| Ours | model_007 | 4 | No | 4 | 50.9 |

Table 7: Comparison of our results with the original paper for the COCO-Stuff-3 dataset.

## 4.4 Performance on the COCO-Stuff dataset

A comparison of our results with the reported pixel accuracy on the COCO-Stuff dataset is shown in figure 8. Our model achieves 6% lower pixel accuracy than the reported value.

| Source | | Model Description | | | Performance |
|---|---|---|---|---|---|
| Reference | Location | Num. res. layers | Attention | Output stride | Pixel Accuracy (%) |
| Original paper | Table 3, col. 2 | 5 | No | 4 | 30.8 |
| Reference | Experiment No. | Num. res. layers | Attention | Output stride | Pixel Accuracy (%) |
| Ours | model_008 | 4 | No | 4 | 24.8 |

Table 8: Comparison of our results with the original paper for the COCO-Stuff dataset.

## 5 Discussion

Due to the changes we made to the model architecture in order to accommodate our available hardware, it is unsurprising that our results indicate generally lower performance than those in the original paper. However we were still able to verify the basic functionality of the method, and our experiments reveal the effects of downscaling the model for use in resource-limited environments. We will proceed by discussing our findings for each of the claims presented in section 2.

Claims 1 and 2 regard the performance of the model on the Potsdam-3 and Potsdam datasets, with the best models achieving reported pixel accuracies of 66.5% and 47.9%, respectively. We reproduced these values to within 1% for Potsdam-3 and to within 4% for Potsdam, therefore broadly supporting the claims. Our experiments on Potsdam and Potsdam-3 also support claim 5, showing that the long-range dependencies enabled by an attention layer can help improve model performance. In the original paper, the authors report only a minor improvement on Potsdam and Potsdam-3 from reducing the output stride from 4 to 2. In contrast, we find that an output stride of 2 performs significantly better than an output stride of 4 in both cases. Due to computational limitations we did not test a model with both an attention layer and an output stride of 2, however we expect that this configuration could be used to further increase our model's performance.

Claims 3 and 4 relate to the model performance on the COCO-Stuff-3 and COCO-Stuff datasets. The authors claim state-of-the-art pixel accuracy among unsupervised methods on these datasets, with 72.9% on COCO-Stuff-3 and 30.8% on COCO-Stuff. We were unable to reproduce these claims using our model, attaining a drastically lower accuracy of 50.9% on COCO-Stuff-3 and an accuracy of 24.8% on COCO-Stuff. This result does not necessarily refute the stated claims, given the simplified model and smaller batch sizes used in our experiments. However, this reveals an interesting contrast with the Potsdam and Potsdam-3 datasets, where our down-scaled model approaches the authors' original results. One possible reason for this is the differences in content between the datasets. The Potsdam scenes are largely uniform in terms of scene geometry, subject matter, and illumination, whereas the COCO-Stuff dataset contains images with features across a wide variety scales, subject matter, and illumination conditions. A larger model may be required to learn the semantic features necessary for obtaining good performance on these benchmarks.

Some examples of our model's performance on Potsdam-3 and COCO-Stuff-3 are shown in figures 3 and 4. It can be seen that the model trained on Potsdam-3 can successfully separate vegetation and trees from the other classes, however it has difficulty distinguishing buildings from roads in some circumstances. The largest issue with the model's performance in the COCO-Stuff-3 dataset is its inability to preserve the edges in the image, resulting in a significant

loss of detail. The examples presented in the original paper do not suffer from this difficulty. It is unclear how the differences between the authors' method and ours account for this.

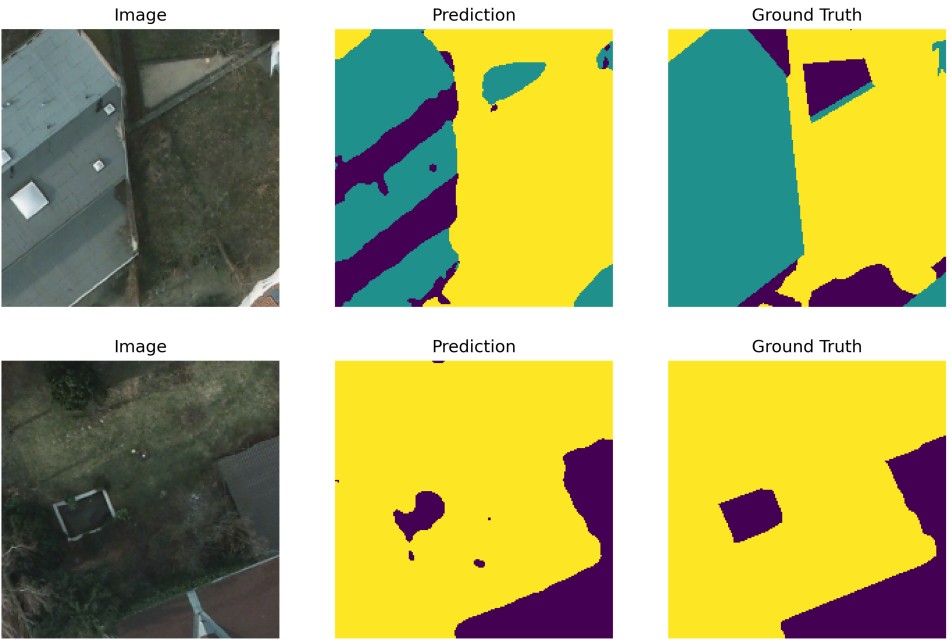

Figure 3: Example segmentations on COCO-Stuff-3 obtained using model_006.

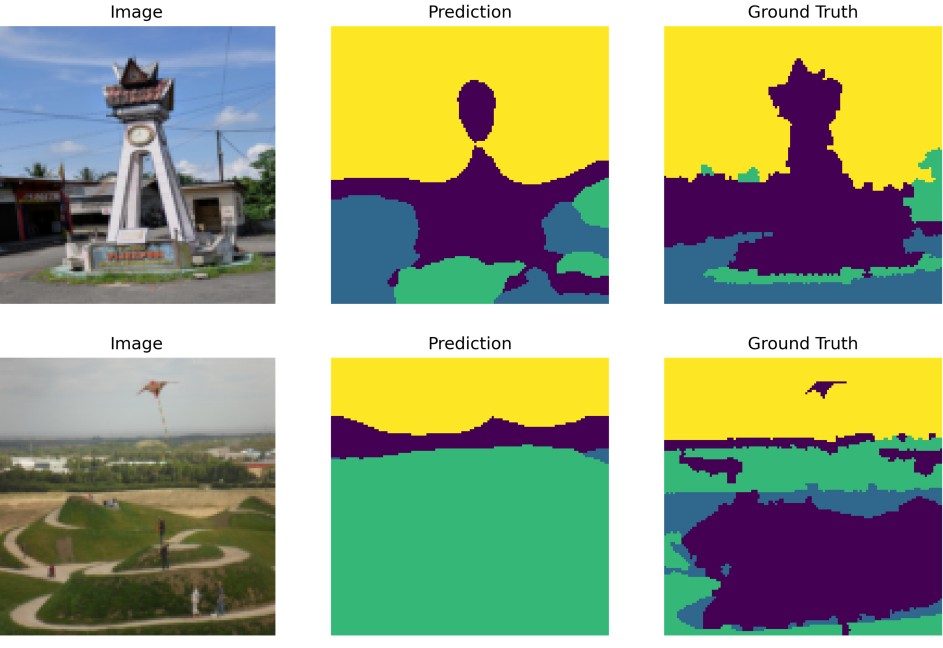

Figure 4: Example segmentations on COCO-Stuff-3 obtained using model_007.

### 5.1 What was easy

Implementing the basic model architecture was straight forward. The masked convolution blocks are easily implemented using standard machine learning frameworks such as PyTorch. Furthermore, masked convolutions can be adopted into essentially any CNN architecture, which opens up possibilities to a wide range of further experimentation and extension of the results in the original paper.

While the authors of the paper did not describe the implementation of the loss function in detail, a related paper which uses the same loss function had code available, which was easily adapted for our experiments.

### 5.2 What was difficult

Our major difficulty during this project was related to limited computational resources. The experiments presented in the original paper were conducted using high-end GPUs, whereas we only had access to a single consumer-grade device. Despite this, we have shown that the model can be scaled down to fit on more limited hardware, while still achieving good results. Our results are therefore relevant for researchers interested in applying unsupervised segmentation techniques in resource-constrained environments or developing more efficient models for this task.

From an implementation point of view, the self-attention mechanism was the most challenging aspect of the model. Self-attention applied to image processing is a new area of research and there are relatively few resources available to help understand it. Additionally, it has high computational complexity, making experimentation difficult.

### 5.3 Communication with original authors

The original paper did not specify the location of the self-attention layer within the model architecture, nor did it specify the sizes $d$ of the projection layers. The original authors were very helpful in clarifying these points by email correspondence. They provided the following details:

1. The self-attention layer is placed within the autoregressive encoder section of the network, after the first two residual blocks.
2. The projection layers have half the number of channels as the inputs. For the architecture in the paper, the channel dimension of the input to the attention layer is 256, so the dimension $d$ of the projections is 128.

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
