# OpenReview forum: "Autoregressive Unsupervised Image Segmentation"
_ML_Reproducibility_Challenge/2020 — Reject_

### Official Review · AnonReviewer1 · 2021-02-28
**Good attempt with clear limitations**

**Rating:** 7
**Confidence:** 5

**Review:**

Overall the authors have done a very good job on reproducing the ECCV 2020 paper "Autoregressive Unsupervised Image Segmentation". As stated in the report, it was relatively straightforward implementing the architecture described in the original paper with only two hurdles; one relating to the loss function, which since it was taken from a previous paper, the authors were able to use information from that paper to reproduce the ECCV paper; second relating some issues around the attention layer, for which they contacted the authors, hence resolving it.

All the steps undertaken are explained concisely and appropriately, including what was easy and what wasn't. Authors managed to approximate results in one of the two datasets used in the original paper. Bottomline is that all the results presented do not match those of the original paper - more or less. The reason is that the authors were not able to use the same resources as the original paper, leading to different batch size and also architecture (4 vs 5 layers). Such changes can have an unpredictable effect to the final results therefore I am not sure one can be certain that the results obtained match - or not - the ones presented in the original paper.

Having said that, there is only so much that can be done if the resources are not available; nonetheless the results are affected.

Overall it is a very good effort and the authors have done a great job in reproducing the paper. It seems that the original paper has had enough information, allowing a relatively straightforward reproduction with two caveats; one being having to contact the authors; and second resorting to a previously published paper to get some information on the loss function.

**Familiar With The Original Paper:**

I have read the original paper

**Reproducibility Summary:**

Report has summary

---

### Official Review · AnonReviewer2 · 2021-03-03
**Good reproduction of smaller-scale results**

**Rating:** 7
**Confidence:** 3

**Review:**

**Strengths:**

1. Further hyperparameter investigation on the Potsdam dataset.

The reproduction does some further hyperparameter scans on the batch size and output stride. They show how some of the accuracy lost from other necessary changes can be regained with some tweaks. This is a good contribution to increasing data on how well the underlying segmentation method works.

2. Good discussion of implications of the experiments on reproducibility success.

While there are some gaps in the completeness of the reproduction, as in the following, these are for the most part clearly described by the submission. Section 2.1 lists specifically which experimental results were replicated. The fact that unsupported claims are possibly unsupported due to differences in the reproduction are clearly described in other parts of the submission.

3. The submission clarifies details in the original significantly.

Some of these are specifically listed in the Section 5.3 (describing the communications with the authors). Architecture specifics such as the number of residual blocks and strides are also given in a clearer and more complete way in this submission than the original paper.

The original paper has some references to supplementary material that no longer seem to be available in the final published form, as far as I could find. Based on the references, some of these details may have been in that document.

This submission collects (from the code, discussion with authors, and possibly other sources) and describes some of them in a clear way that looks like it'd be a really good resource for any reproduction or follow-up work on the original paper.

**Weaknesses:**

4. Reproduction trains fewer epochs for reasons not described.

The command-line arguments in: \
https://github.com/Max-Manning/autoregunsupseg/blob/master/run_experiments.sh \
seem to specify 20 epochs for all trainings.

Section 3.4 of the repro says: "All experiments were run for 10 epochs."

This discrepancy and its effects isn't discussed further, unless I've missed it. Some clarification in the rebuttal may help.

5. Experiments on COCO-stuff are on a smaller model.

The reproduction reduces the number of residual blocks in the auto-encoder. This is done due to limitations on available compute resources.

Hard to draw any conclusion at all from the smaller model on COCO-Stuff. The observed drop in accuracy could be from either the smaller model, or a failure to reproduce. Results on Potsdam with the smaller model do not disambiguate between the two, as the tradeoff for model size vs accuracy can differ between datasets.

The authors of the repro acknowledge this and explain well how it affect their conclusions, so this is a not a strong negative for the score.

It might help to note the comparison to baseline accuracy numbers. E.g. the comparison to other unsupervised ssemantic segmentation methods in Table 3 of the original paper.

**Familiar With The Original Paper:**

I have read the original paper

**Reproducibility Summary:**

Report has summary

---

### Official Review · AnonReviewer3 · 2021-03-04
**Reproduces on a simpler dataset, lacks experiments for the complex one**

**Rating:** 6
**Confidence:** 3

**Review:**

Positives:
1. The authors implement the method using pre-existing well tested code released for a previous paper.
2. The authors are able to reproduce the results on Postdam dataset to within 1%.

Negatives:
1. The authors take a rather narrow outlook in reproducing the paper. Due to computational limits, they reduce the batch size and use a smaller model which makes verifying the claims rather difficult. The authors report that they "do not find evidence supporting this claim..". It should be mentioned that due to computational constraints, they were unable to run the full model in this summary line itself.
2. The authors apply their judgement and refute the claims made by the original paper. They mention that their models couldn't differentiate edges well. But, this could very well be due to a training issue or just to architectural simplifications. It would have been better if the authors also communicated with the original authors to determine the reason for the wide gap in performance on COCO-Stuff.
3. The authors could have summarised the original paper better in Introduction section.

**Familiar With The Original Paper:**

I have read the original paper

**Reproducibility Summary:**

Report has summary

---

### Decision · Program_Chairs · 2021-03-31

**Decision:**

Reject

**Comment:**

Overall reviews and/or the paper content not good enough for the AC to recommend to the journal.